

# Automatic detection of the Earth Bow Shock and Magnetopause from in-situ data with machine learning

Gautier Nguyen[1], Nicolas Aunai[1], Bayane Michotte de Welle[1], Alexis Jeandet[1], and Dominique Fontaine[1]

[1]CNRS, Ecole polytechnique, Sorbonne Université, Univ Paris Sud, Observatoire de Paris, Institut Polytechnique de Paris, Université Paris-Saclay, PSL Research Univsersity, Laboratoire de Physique des Plasmas, Palaiseau, France

**Correspondence:** Gautier Nguyen (gautier.nguyen@lpp.polytechnique.fr)

**Abstract.** We provide an automatic classification method of the three near-Earth regions, the magnetosphere, the magnetosheath and the solar wind in the streaming in-situ data measurement that outperforms the previous methods of automatic region classification. The method was used to identify 14186 magnetopause crossings and 16192 bow shock crossings in the data of 10 different spacecrafts of the THEMIS, ARTEMIS, Cluster and Double Star missions and for a total of 79 cumu-

lated years. These multi-missions catalogs are non ambiguous and can be automatically enlarged with the increasing quantity of data and their elaboration paves the way for additional massive statistical analysis of the two near-Earth boundaries. The development of these algorithms is a promising step towards their usage for the onboard selection of data of interest.

## 1 Introduction

At the first order, the magnetopause is the boundary on which magnetospheric and magnetosheath pressure balance. It acts like

an obstacle for the upcoming supersonic solar wind. It is then located downstream a collisionless bow shock (Burgess, 1995) across which the solar wind becomes subsonic.

The magnetopause and the bow shock delimitates the near-Earth environment into three distinct regions: the magnetosphere, the magnetosheath and the solar wind.

By definition, the shape, location and properties of these boundaries do depend on the upstream solar wind conditions

(Fairfield, 1971). The task of modelling the two boundaries is not new and numerous are the existing magnetopause (Sibeck et al. (1991); Shue et al. (1997); Lin et al. (2010) and references therein) and bow shock (Jeřáb et al. (2005); Farris and Russell (1994) and references therein) analytical, or empirical models (Wang et al., 2013) that provide a decent adequation to the in-situ observation provided by the spacecrafts from various missions.

The ever-growing quantity of near-Earth in-situ data, also allowed statistical studies of the properties of both magnetopause

(Paschmann et al., 2018) and bow shock (Kruparova et al., 2019).

The first step of both empirical modelling and statistical studies is always the same: establishing a consistent catalog of boundary crossing from the streaming in-situ data provided by missions of interest. This appears to be a time-consuming, ambiguous and poorly reproducible task that would be worth making automatic. Using the data provided by the five THEMIS





spacecraft coupled with the solar wind conditions provided by WIND, Jelínek et al. (2012) established a threshold based
method to identify the three near-Earth regions and eventually build crossings lists from this classification but the efficiency of
such a method is unknown and it is not sure how this method could be adapted to the data of other missions especially the one
having a non-equatorial or nightside orbit. Additionally, finding the perfect set of features and the optimal associated thresholds
can appear to also be time-consuming and not flexible enough with regard to the large variability expected from the data.

A way to improve this solution would stand in the use of supervised machine learning algorithms that have the advantage
of rapidly finding the intrinsic differences between different labeled points in a dataset. The use of these algorithms to classify
time series into several categories is not new in the field of space physics. They have especially made their proof to classify the
solar wind into several categories (Camporeale et al., 2017) or to determine if an interval of data actually did contained a Flux
Transfer Event (FTE) (Karimabadi et al., 2009).

In this paper, we use the magnetic field and plasma moments of the THEMIS mission to establish a fast and reproducible
automatic detection of the three near-Earth regions. After a presentation of the data and the algorithm we have been using, we
adapt the method to the data of 3 different missions: Cluster, Double Star and Artemis. We compare the performance of our
method to a threshold based method for the different missions. We finally use our method to automatically elaborate boundary
crossings catalogs.

## 2 Automatic detection of the three near-Earth regions

### 2.1 Data and label

We used plasma moments and magnetic field data from the THEMIS B spacecraft for the whole mission period (between April
2007 and October 2009) during the dayside, dawn and dusk operation phases. The magnetic field data were provided by the
Fluxgate Magnetometer (Auster et al., 2008) with a temporal resolution of 3s. Concerning the plasma moments, we used the
reduced mode of the data provided by the electrostatic analyzer (ESA, McFadden et al. (2008)). The gaps in the data were
filled with the plasma onboard moments (MOM, McFadden et al. (2008)) and the full mode of ESA. The remaining gaps were
filled by time interpolation in order to have regular streaming data through the whole 2007-2009 period. Due to the important
differences existing between the different missions in the specificities of the distribution functions and particle energy or pitch
angle spectrograms, we chose not to consider those products and focused on moments only.

The obtained dataset then consists in 8 input variables: the ion bulk velocity components, $V_x, V_y, V_z$, the magnetic field and
its components, $B_x, B_y, B_z$, the ion density $N_p$ and the temperature $T$. The data was then resampled to a 1 minute resolution.

Each time step was then associated with a label indicating the region in which spacecraft was: 0 for the magnetosphere, 1 for
the magnetosheath and 2 for the solar wind. To avoid manual labelling of each regions over the whole dataset, we initiated our
process with 2 hours of each regions. We then increased our dataset and our labels with the successive, eventually corrected,
predictions made by our algorithm on our training set. The typical labelling of the three regions is shown on the last subplot of
Figure 1 where the theoretical label, shown in blue has been shifted for visual purpose. Following this process, our dataset is
made of 59798 points of magnetosphere, 48056 points of magnetosheath and 150415 points of solar wind.

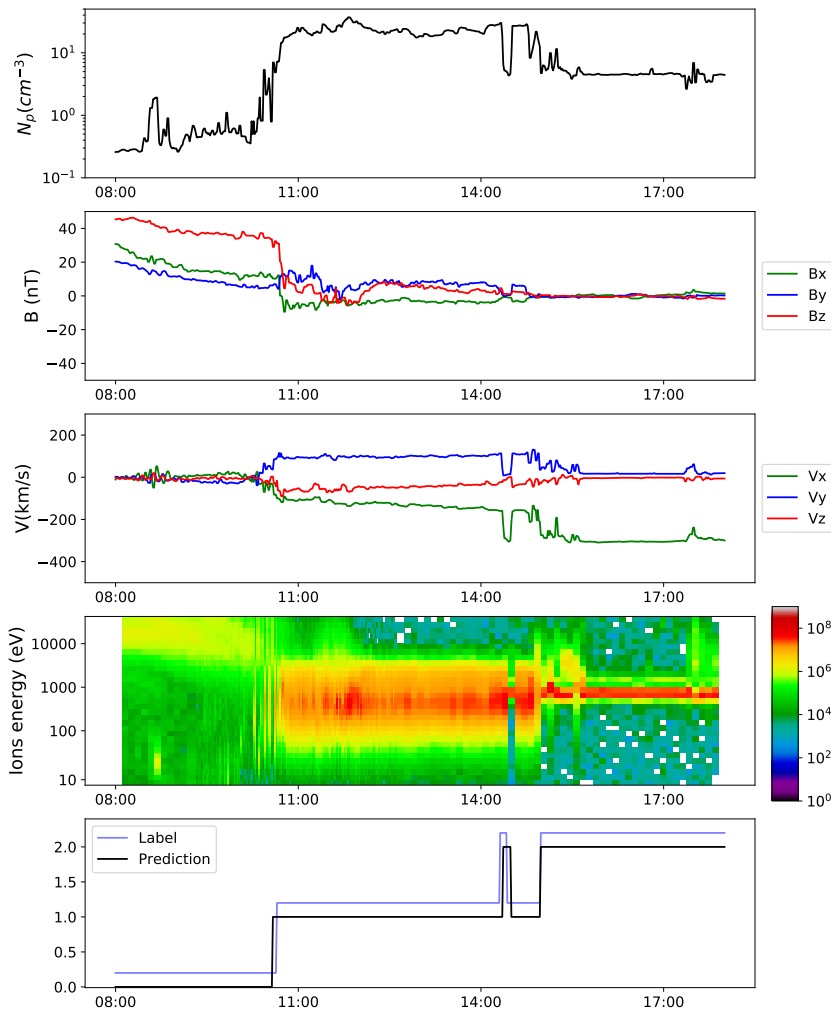

**Figure 1.** In-situ measurement provided by THEMIS B spacecraft on the $12^{th}$ of May 2008. From the top to the bottom are represented: the ion density, the magnetic field components, the velocity components the omnidirectional differential energy fluxes of ions. The last bottom panel represents the evolution of the label (blue) , intentionally shifted for visual inspection and the prediction made by our algorithm (black).



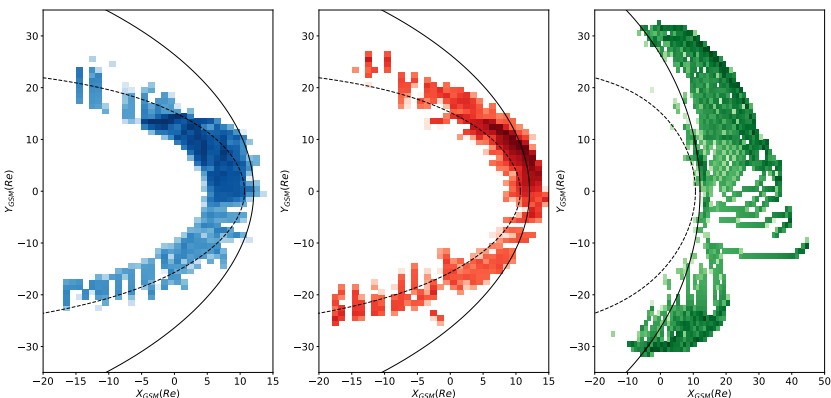

**Figure 2.** Spatial coverage of our labelled dataset projected in the (X-Y) GSM plane, the solid black line represent a stand-off position the bow shock following Jeřáb et al. (2005) model while the dotted black line represent the magnetopause model of Lin et al. (2010). Labels are spatially represented in a log-scale 2D histogram. Magnetosphere bins in blue vary between 1 and 901, Magnetosheath bins in red vary between 1 and 1421, solar wind bins in green vary between 1 and 788

We selected data within dawn, dayside and dusk operation phases of THEMIS and thus expect a good coverage of both magnetopause and shock surfaces. This is confirmed by the actual spatial coverage of our labelled dataset shown in Figure 2. We then expect the method to be robust enough to the variability one can find in the data through the three different THEMIS
operation phases.

In the following, we will designate the subset that has been used to fit our algorithm by *training set*. We will designate by *test set* the remaining subset of data that is used to evaluate the performance of our model. For each of the configurations we have been testing our algorithm with, the *training set* represents 70% of the dataset while the *test set* represents the remaining 30% of the dataset. Unless otherwise noted, all the performances reached by the models that we will be presented have been
obtained with a random split in the dataset between the *training set* and the *test set* for which we ensure the class distribution is still respected.

## 2.2   Algorithm

Gradient boosting is a class of machine learning algorithms that can be used for either classification of regression purposes. The principle of the gradient boosting algorithms is as follows: at first, we train a decision tree, another class of machine learning
based on multiple decisions over the values taken by the features of the dataset, by minimizing the false predictions made over the training set. Additional decision trees are iteratively used to model residuals of the predictions obtained from the previous tree, until the maximum of trees or sufficiently small residuals are reached. The final ensemble of trees is used to model the regions and make predictions on yet unseen data points giving as an output, the probability of belonging to each class. The





final prediction will then correspond to the class with the highest probability. A complete description of the principle and the

fitting process of decision trees and gradient boosting methods can be found in Geron (2017).

With the actual size of our *training set*, it takes two minutes for our Gradient Boosting Classifier to train.

## 2.3  Model performance

After fitting our Gradient Boosting model to our *training set*, we evaluate its performance by comparing its prediction on the *test set* with the corresponding labels. The value of the prediction for a given time interval can be shown in the last subplot of

Figure 1. From then on, a prediction made by our model can be split into four categories for each class:

- A true positive (TP) is a point of a class that has predicted correctly

- A true negative (TN) is a point not belonging to the concerned class that has been predicted correctly (e.g a magnetosheath point that has not been predicted as a magnetosphere point in the case of magnetosphere)

- A false negative (FN) is a point of a class that has not been correctly predicted (e.g a Magnetosphere point that has been

predicted as a Magnetosheath point)

- A false positive (FP) is a point not belonging to the concerned class that has been predicted as belonging to the class (e.g a Magnetosheath point that has been predicted as a Magnetosphere point in the Magnetosphere case)

With these four categories, we can define the *true positive rate* TPR as the ratio between the number of TPs over the total number of expected positives points:

$$TPR = \frac{N_{TPs}}{N_{TPs} + N_{FNs}} \qquad (1)$$

The *false positive rate* FPR is defined as the ratio between the number of FNs over the total number of expected negative points:

$$FPR = \frac{N_{FPs}}{N_{FPs} + N_{TNs}} \qquad (2)$$

An ideal model would be a model without any FN or FP. In this case, we would then expect the TPR to be equal to 1 and the

FPR to be equal to 0 for the three classes. These two values are obtained for a given decision threshold based on the predicted probability as explained in the previous subsection. Logically, low decision thresholds would imply more points predicted as belonging to a certain class and then raise both FPR and TPR. On the opposite, higher decision thresholds would decrease the number of positive points and thus decrease the FPR and the TPR. The evolution of the TPR as a function of the FPR for continuously varying decision threshold can be represented as the Receiving Operator Curve (ROC) shown in the Figure

3 for the three classes. As expected, we notice an increasing TPR with an increasing FPR. The main interest in this curve stands in the inflexion point that correspond to the best compromise we can find between low FPR and high TPR. We want this point to be as close to the top left of each curve as possible as this would imply a FPR close to 0 and a TPR close to 1. A random classifier would, for each decision threshold, increase the TPR and FPR by the same amount and thus be have a ROC





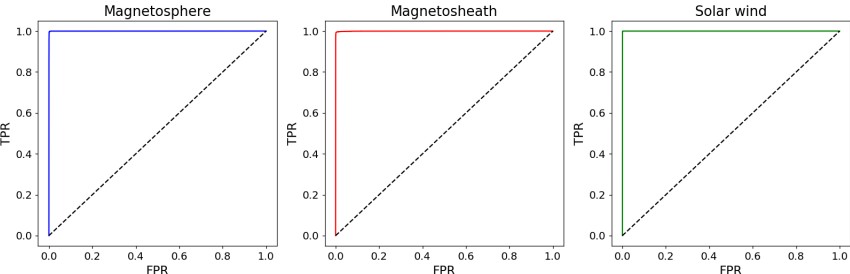

**Figure 3.** ROC curve of our model trained and predicting on THEMIS B data in the case of a random split between the *training set* and the *test set*. From left to right are represented the ROC curve concerning the magnetosphere, the magnetosheath and the solar wind.

representation as a straight line of slope 1. The quality of the ROC can be quantified by by computing the area under curve
(AUC) of each of the ROCs. And we then expect this AUC to be as close to 1 as possible. To ensure the independence of our
method from the split we made between *training* and *test* set, we trained and predicted our model 10 times for 10 different
splits and computed the AUC. The average AUC we obtained for each class is shown on the first row of Table 1. At first, the
high AUC obtained for the three classes indicate how well our model perform in classifying the three regions. Moreover, the
low standard deviation we obtain is lower than $1e-3$. This shows that our method is independent from the split we make
between our two sets.

By performing this random split, we allow temporally close points, and thus almost identical points in the features space to
be in both training and test set. Consequently, performing this split only partially shows the reproducibility of our method and
how well it would perform on really unknown data. To ensure there were not any temporal overfit due to this split and that
our method was truly reproducible, we trained and evaluated our model 3 times by splitting our dataset temporally instead of
randomly. For this, we considered our *training set* to be a time interval that represented $2/3$ of our dataset and left the remain
to be the *training set*. As the average AUC we obtained in this case, also shown in Table 1, has very few variations compared
to the random split, this ensures the temporal independance of our method as well as our reproducibility.

Last but not least, the greatest part of our label has been obtained after successive train and prediction of our model on the
parts of the dataset that had not been labelled yet. Thus, the label can eventually contain errors that could affect the quality
of our prediction and high AUC would then not indicate the classification ability or our model but its ability to learn from an
erroneous label. To figure this out, we led trainings and evaluations of the algorithm by voluntarily mislabelling $25\%$ of our
dataset. If our model completely followed the indicated label in the training set, we would once again expect a high AUC. This
is not the case, as shown in Table 1. This shows the real capacity of our algorithm to classify the three near-earth regions as
well as the reliability of our label.

| Mission | AUC Magnetosphere | AUC Magnetosheath | AUC Solar Wind |
|---|---|---|---|
| THEMIS B (w. Random split) | 0.999 | 0.999 | 0.999 |
| THEMIS B (w. Temporal split) | 0.999 | 0.997 | 0.999 |
| THEMIS B (w. Mislabelling) | 0.75 | 0.707 | 0.752 |
| Cluster 1 (without retraining) | 0.988 | 0.983 | 0.996 |
| Cluster 1 (with retraining) | 0.999 | 0.998 | 0.999 |
| Double Star TC1 (without retraining) | 0.996 | 0.992 | 0.996 |
| Double Star TC1 (with retraining) | 0.999 | 0.998 | 0.999 |
| Artemis | 0.999 | 0.999 | 0.999 |

**Table 1.** Comparison of the Area Under Curve of the ROC curve of our detection algorithms for different missions.

## 3 Adaptability of the model: from a mission to the other

Having developed an automatic detection method of the three near-earth regions with high reliability, one could think of its adaptation to the data provided by additional spacecraft that went through these regions. In this section, we will adapt our method to the data provided by Cluster, Double Star and ARTEMIS.

### 3.1 Cluster

In the case of Cluster, we used the available data from Cluster 1 between the 1st of January 2001 and the 1st of January 2013. The magnetic field data were provided by the Fluxgate Magnetometer with a temporal resolution of 4s (Balogh et al., 2001). The plasma moments were provided by the Hot Ion Analyzer instrument (Rème et al., 2001) when the instrument was working under the magnetosphere or the magnetosheath mode. Following what we did for THEMIS, the data were resampled to a 1 minute resolution.

A typical representation of such data is shown in Figure 4.

To check out the adaptability of the model we trained in the previous section, we labelled 50277 points of magnetosphere, 76468 points of magnetosheath and 22017 of solar wind between the years 2005 and 2006. The spatial distribution of this labelled dataset is shown in the Appendix A.

Intuitively, one could think that the efficient model we have trained in the previous section on THEMIS would also perform well on Cluster data. Nevertheless, the lower AUC we have in Table 1 without retraining indicate the adaptability is not this obvious. This is mostly due to the spatial coverage of THEMIS that has a much more equatorial orbit than Cluster. The data provided by the two missions can therefore be substantially different and an algorithm that has only seen equatorial data could perform better on unseen polar data. To cope with it, we refitted the model trained on THEMIS with a *training set* we picked in our labelled Cluster data. The resulting model was then evaluated on the remaining *test set* and the operation was repeated for 10 different *train-test* split configurations in a similar way than what was done for THEMIS. The increasing AUC we obtained, shown in Table 1, proves the necessity we had to adapt our algorithm to the specificity of the cluster data as well as the capacity

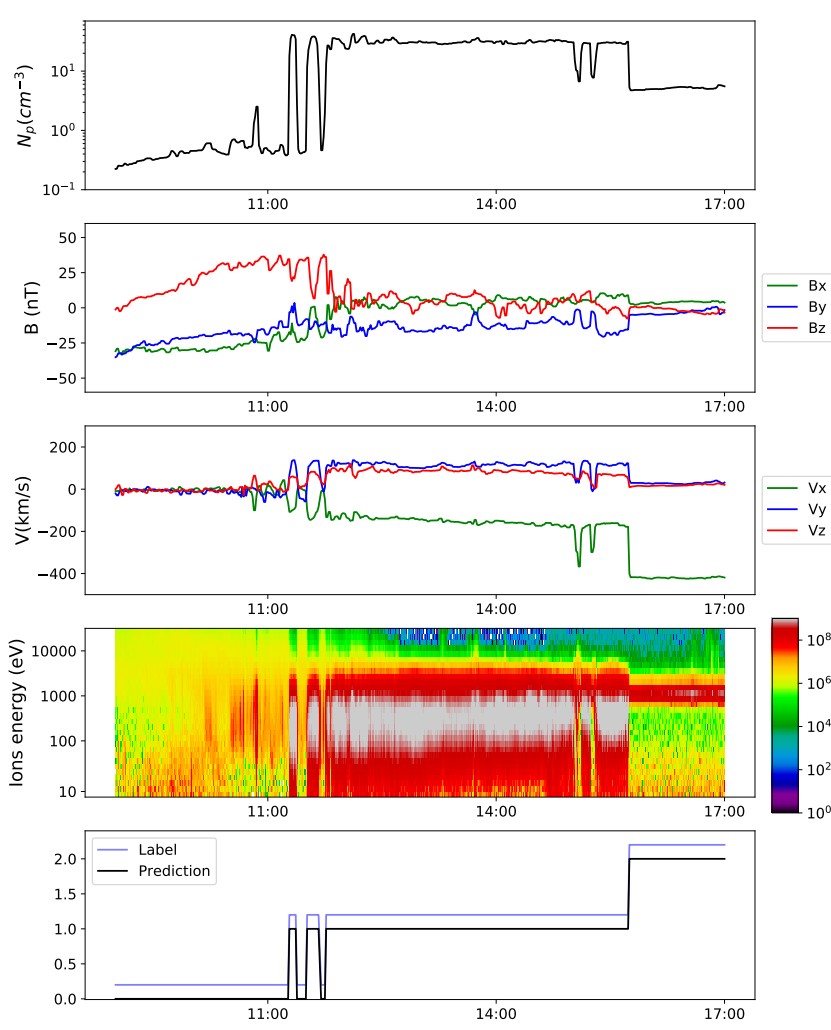

**Figure 4.** In-situ measurement provided by Cluster 1 on the $6^{th}$ of February 2005. The legend is the same than in 1



we have to adapt our algorithm to the data of another mission after a small labelling phase to take into account the mission specificity.

## 3.2 Double Star

In the case of Double Star, we used the data of the TC1 spacecraft whole mission period (between the 1st of January 2004 and the 1st of january 2008) The magnetic field data were provided by the Fluxgate Magnetometer (Carr et al., 2005) with a temporal resolution of 2s. The plasma moments were provided by the CIS-HIA instrument (Fazakerley et al., 2005) with a temporal resolution of 4s. Once again, the data were resampled to a 1 minute resolution.

A typical representation of the data is shown in Figure 5.

Here the part of the data we labelled is made of 20671 magnetosphere points, 23091 magnetosheath points and 4944 solar wind points taken at the beginning of the year 2005. The main reason explaining the noticed imbalance in the data stands in the orbit of TC1 itself that is not supposed to cross the bow shock. The spatial distribution of our labelled data is also shown in Appendix A.

As TC1 also has an equatorial orbit, we expect the model trained on THEMIS to perform well even without having to be 160 retrained and this is the main reason why we do not have an entire coverage of the (Y-Z) plane in the specific case of Double Star. We have this confirmation with the value of the AUC without retraining in Table 1. Refitting the model would then allow a finer detection that would be specific to the quality of the data provided by TC1 in comparison to the THEMIS data and this is indicated with the increasing AUC with a retraining of the model with our labelled data.

## 3.3 Artemis

The mission ARTEMIS actually corresponds to the THEMIS B and C spacecrafts when they moved from a terrestrial to a lunar orbit at the end of the year 2009. The data we used in this case are then the one provided by the same instruments of THEMIS B than the one we mentioned in section 2 between the 1st of January 2010 and the 1st of June 2019. With the change of orbit, it is necessary to take into account the moments in the data for which the spacecraft will be in the moon's wake. This new region will then be considered as an additional possible region (and thus another label with the value 3) for the data. A typical 170 representation of the data that includes a passage in the moon's wake is shown in Figure 6. As, in this case, the spacecraft spends the greatest part of its orbit in the solar wind, and because our dataset consists of a quasi complete solar cycle, we do here expect much more variability of the data specific to this class than what we could have had for the three previous missions. To cope with it, we labelled a month per year of our dataset for a total of 26560 magnetosphere points, 131656 magnetosheath points, 429283 solar wind points and 15070 points of moon's wake which spatial distribution is shown in Appendix A.

Additionally, the greatest part of the data here corresponds to the distant night side for which the transition from the magnetosheath to the solar wind is much less obvious than on the dayside and where small velocity, density and temperature variations could be wrongly interpreted as a boundary crossing. To cope with it, we then added the spacecraft GSM coordinates as a feature of the dataset which will then consist in 11 input variables.



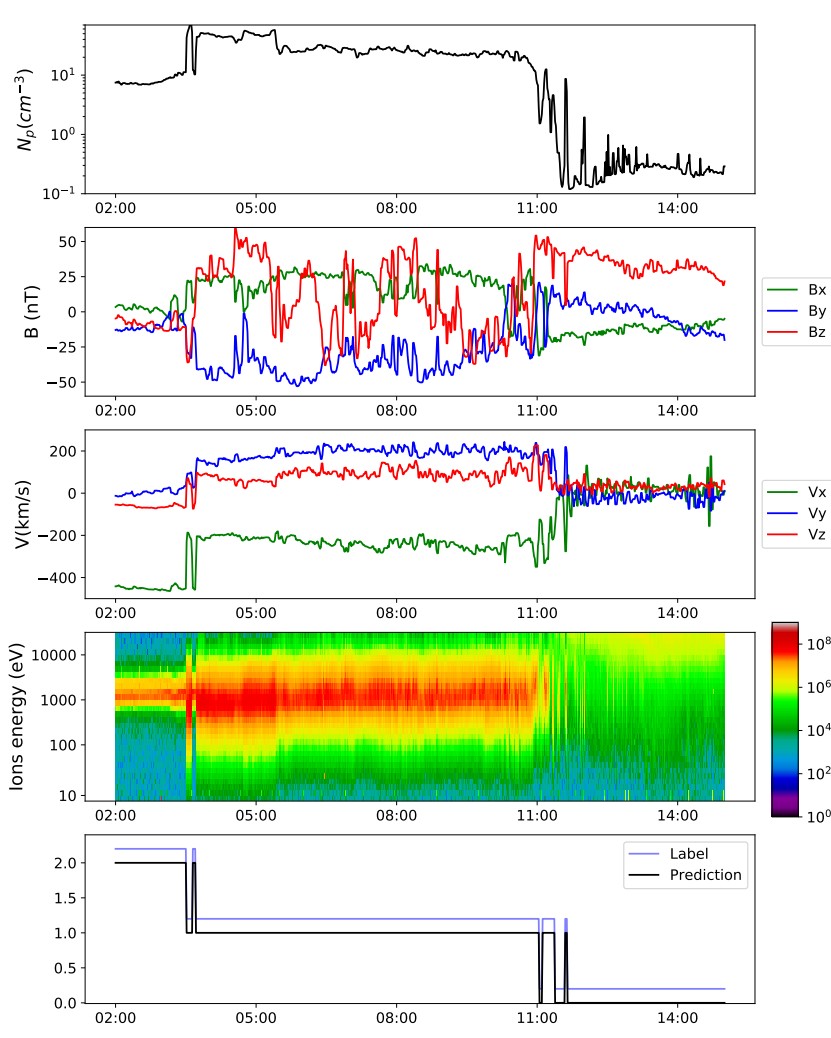

**Figure 5.** In-situ measurement provided by Double Star TC1 spacecraft on the $1^{st}$ of January 2005. The legend is the same than in 1





| Mission | AUC Magnetosphere | AUC Magnetosheath | AUC Solar Wind |
|---|---|---|---|
| THEMIS B | 0.915 | 0.908 | 0.859 |
| Cluster 1 | 0.897 | 0.852 | 0.828 |
| Double Star TC1 | 0.913 | 0.894 | 0.843 |

**Table 2.** Comparison of the Area Under Curve of the ROC curve for the threshold-based method

Having a different dataset and a different number of classes, it is here no use of using the model we trained in the previous
section and we will then focus on the specific model we trained for this mission. The resulting high AUC shown in Table 1
proves the adaptability of this method to another kind of orbit and and its flexibility and robustness regarding the addition of
another region.

## 4   Comparison with threshold based method

The automatic identification of the three near-earth regions had already been attempted by Jelínek et al. (2012) with the use
of thresholds on the magnetic field magnitude and the density. Nevertheless, the quality of the classification has never been
evaluated nor its adaptation to non-equatorial data.

Figure 7 represents the 2D histogram of $B$ and $N_p$ for THEMIS B, TC1 and Cluster 1 on the periods on which we labelled
our different datasets. We divided these parameters by the corresponding solar wind density and the IMF that we obtained from
the shifted OMNI data. At first sight, one can easily distinguish three main regions that are separated with the solid red lines
for the three missions. Nevertheless, these linear boundaries have been set manually and we can not ensure these could the
best choice for the three missions. To evaluate the quality of the classification, we computed the TPR and the FPR for the three
missions and for varying boundary lines. We then used these values to compute the AUC that is shown in the table 2. Once
again, we notice a lower AUC in the case of Cluster which is consistent with the difference we have between equatorial and
polar orbits as explained in the previous section. Additionally, even if the boundaries plotted in the Figure 7 seem to provide
a decent separation between the three regions, the AUC is lower than the one we obtained with the gradient boosting. Which
indicates our model performs better in classifying the three regions by setting more flexible boundaries while requiring less
time than the one required to set the thresholds used in the Figure 7. The same kind of histogram gets messier with a much less
obvious transition from the magnetosheath to the solar wind and the addition of the moon's wake as shown with the Artemis
data in the Appendix C. This shows the difficulty a threshold-based method would have for a night side oriented mission and
the interest of using machine learning in this case.

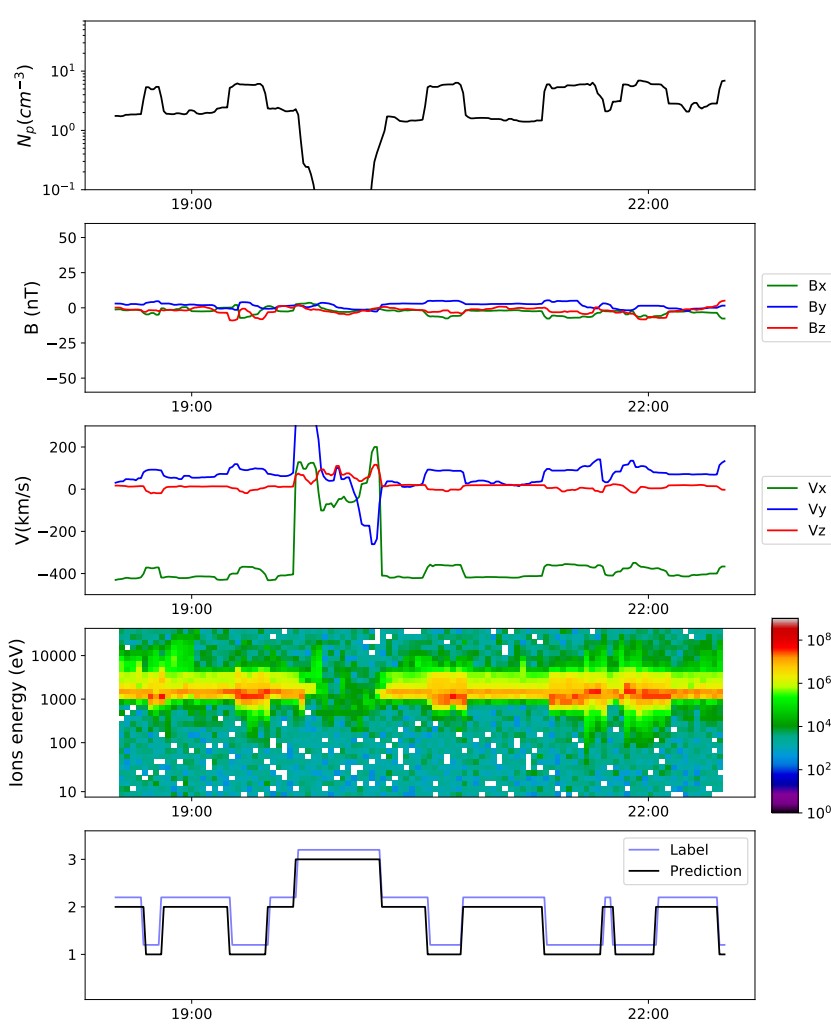

**Figure 6.** In-situ measurement provided by the ARTEMIS 1 spacecraft on the $13^{rd}$ of August 2016. The legend is the same than in 1





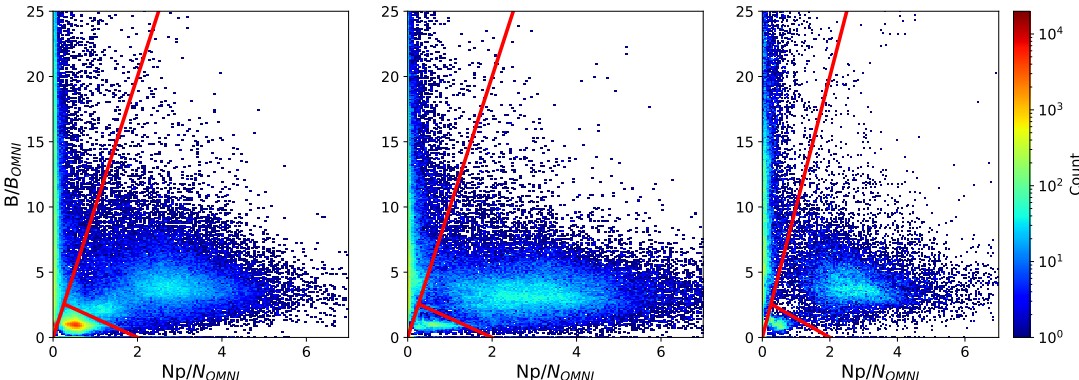

**Figure 7.** 2d histogram of $B$ and $N_p$ divided by the corresponding OMNI data for the three missions: THEMIS B (left), Cluster 1 (middle) and Double Star TC1 (right). The solid red lines indicate a possible set of linear boundaries we could define to separate the three regions

## 5   Automatic elaboration of boundary crossings catalogs

Our method proved its reliability and efficiency in classifying the three near Earth regions. From now on, we could use it to elaborate our own magnetopause and crossing catalog by classifying the streaming in-situ data provided by any near-Earth space craft and by selecting time intervals enclosing two predicted regions.

### 205   5.1   Magnetopause catalog

In the case of the magnetopause, we define a crossing as a 1 hour interval that contains as much magnetosheath points as magnetosphere points. We then elaborate a complete magnetopause crossing by running our model trained in section 2 on the data provided by THEMIS A, B, C, D and E spacecraft. To gain time in the construction of the crossings and because we do not expect any magnetopause crossing in these regions, we restricted ourselves to the dayside, dawn and dusk operation
phase and we removed the parts of the orbit in which the spacecraft was far away in the solar wind ($X_{GSM} > 15Re$). The same process has been made on the in-situ data provided by Cluster 1 on the 2001-2016 period, by Cluster 3 on the 2001-2009 period and on Double Star between 2004 and 2007 by using the corresponding model presented in section 3. The total number of crossings we obtained are summarized in the Table 3. And all of the magnetopause lists can be found here: https://github.com/gautiernguyen/in-situ_Events_lists.

Given that our detection method has been evaluated on large parts of orbits, the high quality of the classification is made with regions where the spacecraft is not expected to cross a boundary. In these regions, the algorithm is less likely to hesitate on its prediction. On the other hand, it is more probable it hesitates on the predictions made close to the boundaries. Consequently, we have to ensure the classification is still of decent quality. Figure 8 represents the ROC we have on the classification between magnetosphere and magnetosheath points for THEMIS B, Cluster 1 and Double Star for the subset of our test set that lies in the
proximity of a magnetopause or shock crossing. These predictions have been obtained with a model that has been trained with



| Mission | Magnetopause crossings | Bow shock crossings |
|---|---|---|
| THEMIS A | 2824 | 1590 |
| THEMIS B | 373 | 1030 |
| THEMIS C | 658 | 1238 |
| THEMIS D | 2691 | 1520 |
| THEMIS E | 2726 | 1511 |
| Cluster 1 | 1813 | 3225 |
| Cluster 3 | 1534 | 2004 |
| Double Star TC1 | 931 | 846 |
| ARTEMIS B | 263 | 1602 |
| ARTEMIS C | 373 | 1626 |
| Total | 14186 | 16192 |

**Table 3.** Number of magnetopause and bow shock crossings we have for different missions

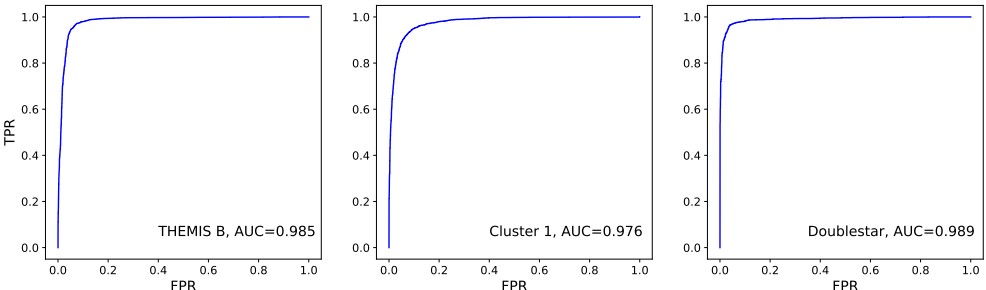

**Figure 8.** ROC curve specific to the magnetosphere evaluated on the labelled crossings for the three missions THEMIS B (left), Cluster 1 (middle) and Double Star (right)

the complement part of the dataset, i.e. the subset that excludes the proximity of the crossings. Even if the AUC is lower than the one we obtained in the previous section, its still high value indicate the good quality of the classification when a spacecraft arrives close to the magnetopause and thus our capacity of building crossings from the prediction made by our model.

Another method we could have to ensure the consistency of the obtained crossings would stand in the certitude of the prediction made by the algorithm and their position in comparison to a standard magnetopause position. To do so, we computed the mean probability of each crossing by averaging the probabilities of belonging to the predicted class of each point present in the crossing. Events with high probability would then correspond to undoubtful crossings while the events with the lowest probability would be the most likely to be actual crossings. The probability distribution of our 14186 is shown in Figure 9. Having a high probability for the greatest part of our events then ensures the consistency of our magnetopause lists.





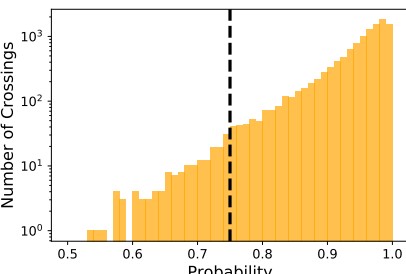

**Figure 9.** Distribution of the probability of the 14186 magnetopause crossings we built and summarized in Table 3. The solid dashed line represent the probability threshold we chose for the Figure 10

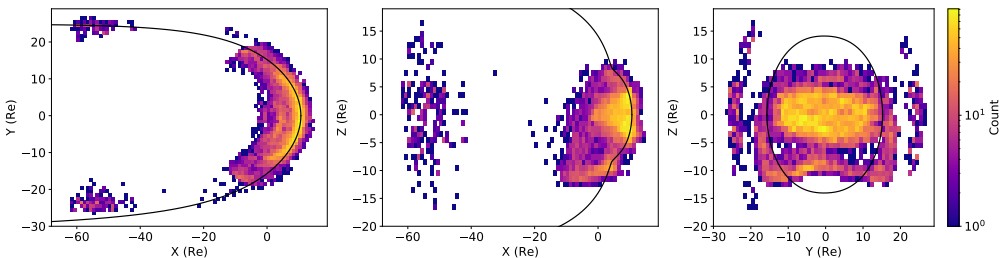

**Figure 10.** Spatial distribution of the crossings above the threshold in Figure 9 in the XY (left), XZ (middle), YZ (right) GSM planes. The solid black line indicate the Lin et al. (2010) magnetopause model with a dynamic pressure of 2 nPa and a null $B_z$.

Finally, the spatial distributions of the crossings that have a probability higher than $75\%$ in the GSM XZ, XY and YZ planes is shown in Figure 10. These crossings represent $98.5\%$ of the crossings built with our models and are then expected to be the most likely to be actual magnetopause crossings. The solid black lines represent the stand off position of the magnetopause model established by (Lin et al., 2010) computed for a dynamic pressure of 2 nPa, a null $B_z$ and assuming no dipole-tilt. The proximity we have between this distance and our actual crossings ends up proving the capacity our method has to elaborate a
decent magnetopause crossings catalog with a decent coverage of the magnetopause at all latitudes and longitudes.

### 5.2    Bow shock catalog

We define a bow shock crossing event as 10 minutes interval that contains as much magnetosheath points as solar wind points. We then run the models we trained for the different missions detailed in Section 3 on the same dataset we used in the case of the elaboration of the Magnetopause crossing catalog. The total number of obtained crossings can is once again summarized
in Table 3 and the bow shock lists can also be found at https://github.com/gautiernguyen/in-situ_Events_lists.

     In a similar way to what has been done for the magnetopause, we ensure the consistency of the massive bow shock crossing detection by evaluating the algorithm on the part of the crossings for which we had labelled data. The associated ROC curve





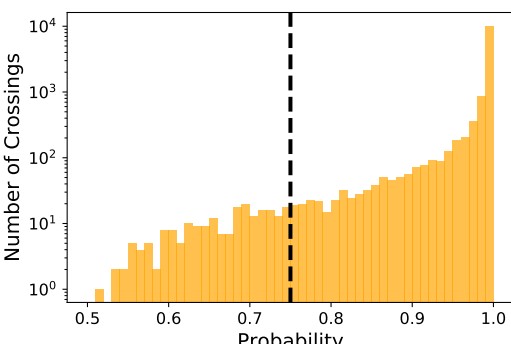

**Figure 11.** Distribution of the probability of the 16192 bow shock crossings we built and summarized in Table 3. The solid dashed line represent the probability threshold we chose for the Figure 12

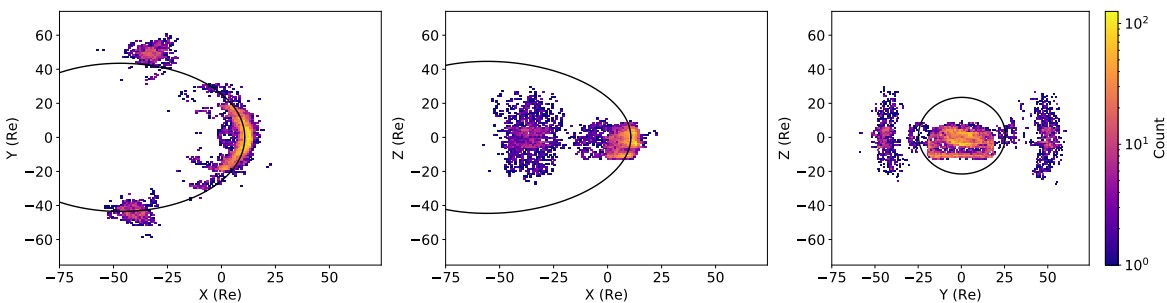

**Figure 12.** Spatial distribution of the crossings above the threshold in Figure 11 in the XY (left), XZ (middle), YZ (right) GSM planes. The solid black line indicate the Jeřáb et al. (2005) bow shock model with a dynamic pressure of 2 nPa, a null $B_z$ and an Alfven Mach of 8.

can be found in the Appendix B and the high value of the AUC coupled to the probability distribution we have in Figure 11 convince us that the greatest part of the detected events are actually bow shock crossings.

The spatial distribution of the crossings with a probability higher than 75% in the GSM XZ, XY and YZ planes is shown in the Figure 12. The solid black line here represents the stand off position of the Jeřáb et al. (2005) bow shock model computed for a dynamic pressure of 2 nPa, a null $B_z$ and an Alfven Mach of 8.

## 6    Discussion and conclusion

Using a Gradient Boosting Classifier, we established an automatic detection method of the different near-Earth environment
regions when they are traversed by the THEMIS spacecrafts during the dawn, dusk and dayside mission phase.

After a small retraining phase, necessary to consider the differences between the data of different missions, this method was adapted to the method to other equatorial (Double Star) and non-equatorial (Cluster) missions. The adaptability of the



method has even been tested on night side oriented missions for which we obtained the same quality of prediction after the addition of the moon's wake as an additional class and the spacecraft position as an additional feature. Having proved this adaptability, we could also think of using the method on the data of additional near-Earth missions such as MMS, IMP8 or ISEE. The classification could even be enhanced by the consideration of additional regions like the Ion foreshock. Such light-weight algorithms could eventually be taken onboard of upcoming missions to automatically select the data of interest and thus automatically decide of the data that should be stored and kept for further analysis. Moreover, the method does not use the specificity of being in the near-Earth and could then also be adapted to other planetary missions in the solar system.

We used this method to elaborate one of the largest existing magnetopause and bow shock crossings catalogs that will be automatically enlarged with the increasing quantity of data. Having a large list of events also gives the opportunity to study these two near-Earth boundaries and physical processes occurring in their vicinity, from a statistical point of view . One could think, for instance, of the identification of the magnetic reconnection jets, which will be the topic of a forthcoming study.

If the work has been done for the magnetopause and the bow shock, the same process presented in Section 5 could be used to build a moon's wake catalog which would pave the way for its study from a statistical point of view.

Last but not least, the high number of events we found is expected to be linked with a great variety in the associated solar wind conditions and it could then be interesting to link the position of the crossings with these upstream conditions through the construction of magnetopause and bow shock data-driven models.

*Code and data availability.* THEMIS data are accessible via the NASA Coordinated Data Analysis web (https://cdaweb.sci.gsfc.nasa.gov/index.html/). Cluster and Double Star data are accessible via the Cluster and Double Star Science archive (http://csa.esac.esa.int/). All of our trained algorithms and labelled dataset can be found here https://github.com/gautiernguyen/in-situ_Events_lists

## Appendix A: Spatial coverage of the labelled datasets

The spatial coverage of the three missions, Cluster, Double Star and Artemis is shown in Figure A1. The spatial position of different passages in the Moon's wake we labelled in the case of Artemis is shown in the Figure A2

## Appendix B: ROC curve on the labelled crossings for the magnetosheath and the solar wind

## Appendix C: 2D histogram of $B$ and $N_p$ in the case of Artemis

*Competing interests.* The authors declare that they have no conflict of interest.



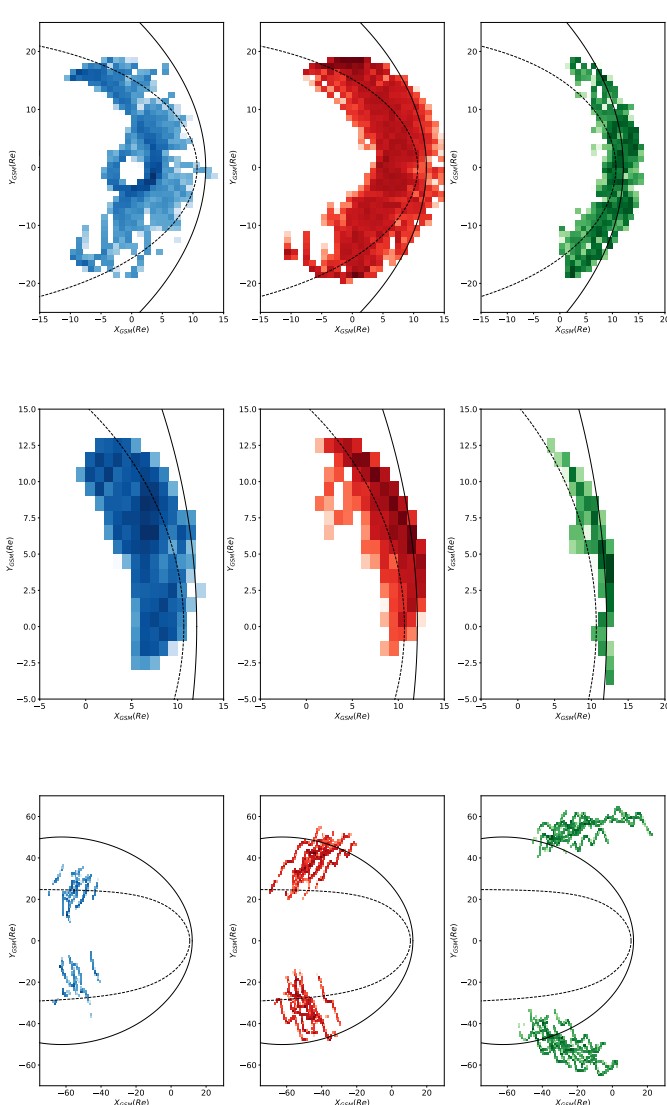

**Figure A1.** Spatial coverage of the labelled dataset of the other missions: Cluster (*Top*), Double Star (*Middle*) and Artemis (*Bottom*). The legend is the same than in Figure 2 with the addition of the passages in the Moon's wake in the case of Artemis that are represnted in Cyan.




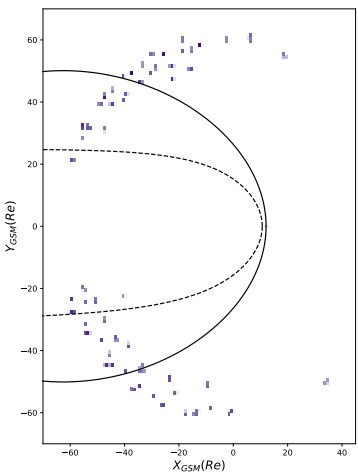

**Figure A2.** Spatial position of the labelled passages in the Moon's wake for Artemis. The legend is the same than in Figure 2. The Moon's wake bins vary between 1 and 157

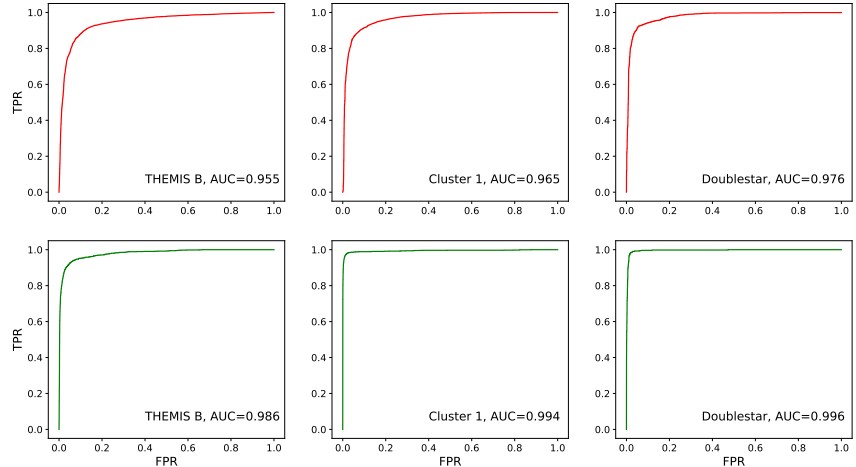

**Figure B1.** ROC curve specific to the magnetosheath (red) and to the solar wind (green) evaluated on the labelled crossings for the three missions THEMIS B (left), Cluster 1 (middle) and Double Star (right)





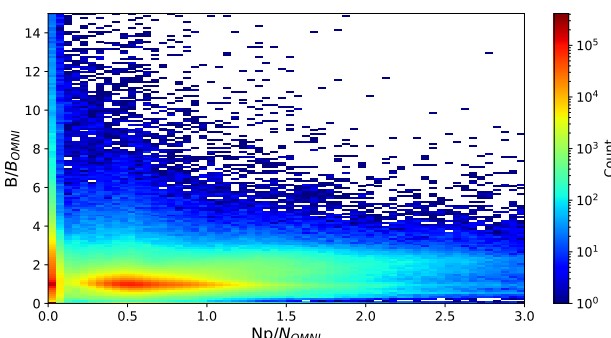

**Figure C1.** 2d histogram of $B$ and $N_p$ divided by the corresponding OMNI data for Artemis B

*Acknowledgements.* While submitting this paper, we came aware that Olshvesky et al. have also discussed how to detect plasma regions in the data of MMS using a 3D Convolutional Neural Network Olshevsky et al. (2019).





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
