# Peer review of "Automatic detection of the Earth Bow Shock and Magnetopause from in-situ data with machine learning"

_Annales Geophysicae, 2019_

## Referee Comment (RC1) · Anonymous Referee #1 · 26 Nov 2019

This paper presents an application of machine learning for the automatic classification of satellite data, specifically to distinguish between magnetosphere, magnetosheath and solar wind regions. The paper is interesting and the machine learning approach is novel. However, there are several flaws that need to be fixed before publication.

1) The authors seem to be aware that a random splitting of the data between training and test set is yields erroneous scores (line 111). Yet, they still present results based on random split. I suggest to completely remove the results obtained in this way and to present only the results obtained with a more correct split in time.

2) The labeling of the data is completely unclear. Reading from line 53 it seems that

they are mixing the 'ground truth' with the result of the classification algorithm. The same argument is repeated on line 118. Obviously you cannot use the same algorithm to label and predict.

3) Similarly, on line 134 they do not explain how do they label such large amount of Cluster data. Similar for other data.

4) In evaluating the model performance, the authors focus exclusively on the AUC. I suggest to compute and show also the True Skill Score and the Heidke Skill Score, that are standard skill scores.

5) Figure 3 is not very informative. It looks like the model can achieve perfect predictions?

6) Line 75: a reference is needed to the original works on boosting algorithms. 7) Line 76: the compute time depends on the CPU (and/or GPU)

8) Line 81: should be 'has been predicted'

9) Line 197: why a decision tree should require less time than an arbitrary decision boundary?

10) Line 199: Decision trees are also threshold-based methods. I do not understand this distinction.

11) Line 203: don't you have cross-calibration issues when you employ the algorithm trained on one satellite to make inference on another satellite?

12) Since decision trees are easily interpretable, it would be interesting to visualize the boundaries and understand their physical implications.

13) Line 226: it is well-known that the probabilities output of boosted ensemble of decision trees are not well-calibrated.

14) Line 275: Do Appendixes B and C really have no text?

15) Finally I suggest to be more specific in the title, by replacing 'Machine Learning' with 'Decision Trees'
* * *

---

## Referee Comment (RC2) · Anonymous Referee #1 · 28 Nov 2019

I thank the authors for answering my questions. Unfortunately, it is now clear that what I suspected were a major flaw of this paper is indeed an obvious mistake in the methodology, which explains the almost perfect prediction accuracy reported. In my opinion, this paper has no value and should be rejected.

The problem is with the automatic labeling of data, starting with a minuscule portion of labeled data. This is in essence what you are doing (of course, correct me if I am wrong). You manually label a very small portion of the data («1%). The decision tree partitions your 8-dimensional input space in a given way. Each partition (8-D hypercube) is then assigned to the majority class (the class that has the majority of

training points in that portion, gets it all. Note that by starting with 6 hours = 360 points in 8 dimensions, you have about 2 points per dimension!). You will agree that by no means this is expected to be a good classifier for the rest of the data. Then you take some unlabeled data, classify them with this decision tree and put them in the training set. If you now train another decision tree with the new enlarged training set, what do you expect? The new classifier will be pretty much identical to the old one, but now each partition will have a larger majority class (because the new training points you have added all belong to the majority class). So the classifier is more confident. You repeat this procedure a number of times, sure enough each leave of the decision tree ends up having a ratio of positive/negative close to 100% or 0%. You have cooked up a perfect classifier (ie perfect with respect to the labels that it has itself produced!)

This is nonsensical.

If you don't have labeled data you are better off with an unsupervised technique.

In any case, I urge you to collaborate with a machine learning expert, to avoid wasting your time with other silly mistakes. Good luck!

---

## Author Comment (AC1) · 28 Nov 2019

We thank the referee for the numerous comments regarding our methodology and the various suggestions proposed to better our work. Please find below the detailed answers to the referee's questions and comments.

**1) The authors seem to be aware that a random splitting of the data between training and test set is yields erroneous scores (line 111). Yet, they still present results based on random split. I suggest to completely remove the results obtained in this way and to present only the results obtained with a more correct split in time.**

[Figure]

As shown in Table 1, we obtain the same score for both random and temporal split. Thus, we do not expect erroneous scores on the prediction even with a random split. We prove it by attaching to our answer the ROC curve we obtain for THEMIS in the case of a temporal split. This attached figure could eventually replace our Figure 3. Additionally, the whole labelled dataset is used to train the final model we use for the massive detection of magnetopause and bow shock crossings. There is then no risk of erroneous predictions due to this random split in the framework of the massive detection. We are aware that this specificity is not specified yet and this shall be the case in a revised version of the paper.

**2) The labeling of the data is completely unclear. Reading from line 53 it seems that they are mixing the 'ground truth' with the result of the classification algorithm. The same argument is repeated on line 118. Obviously you cannot use the same algorithm to label and predict.**

The explanation provided in the paper is a bit unclear and shall be modified in the revised version of the paper. We started our work with 2 continuous hours of each of the 3 regions and trained a first algorithm with this 6 cumulated hours of data. We then augmented the quantity of labelled data by adding the predictions of this first model on additional data intervals with a possible manual correction of the mis-predicted points. A second model was then trained with this enlarged labelled dataset. The operation was used to speed up the labelling process, and was repeated until we reach the scores that are shown in the paper.

**3) Similarly, on line 134 they do not explain how do they label such large amount of Cluster data. Similar for other data.**

We applied the same process as described above for every missions. This point shall be specified in a revised version of the paper.

**4) In evaluating the model performance, the authors focus exclusively on the AUC. I suggest to compute and show also the True Skill Score and the Heidke**

**Skill Score, that are standard skill scores.**

The TSS and the HSS both apply to a specific threshold/cutoff (and thus to a specific confusion matrix) while the ROC curve and associated AUC apply to the full range of thresholds. The information brought by these two metrics is then already shown with the Figures 3 and 8 for every decision threshold.

**5) Figure 3 is not very informative. It looks like the model can achieve perfect predictions?**

The model can indeed achieve almost perfect predictions (the AUC would have been 1 in the prefect case) but still make errors especially when a spacecraft comes across the boundary between two distincts regions. This is what is shown in Figure 8 and explained in section 5.1.

**6) Line 75: a reference is needed to the original works on boosting algorithms. 7) Line 76: the compute time depends on the CPU (and/or GPU)** We agree with the referee that an original reference and the characteristics of the CPU we used will be specified in the paper. For instance, we used an AMD ryzen threadripper 2990wx processor.

**8) Line 81: should be 'has been predicted'**

This will be modified in a revised version of the paper

**9) Line 197: why a decision tree should require less time than an arbitrary decision boundary?** We agree with the referee that the prediction time will be unchanged whether it be a boosted ensemble of decision trees or an arbitrary decision boundary. The difference stands in the fitting time of the Gradient Boosting compared to the time required to define the arbitrary boundary that provides the best output. We shall make this point clearer in the revision.

**10) Line 199: Decision trees are also threshold-based methods. I do not understand this distinction.**

By "threshold-based methods", we mean "manually-set thresholds" that do not have the ability to go as much into precise decision steps as Decision trees and that are not based on sound statistical properties of the dataset but rather on empirical subjective knowledge of the operator. We agree with the referee that the distinction as mentioned in the paper is a bit unclear and shall be emphasized by the addition of the mention "manually set".

**11) Line 203: don't you have cross-calibration issues when you employ the algorithm trained on one satellite to make inference on another satellite?**

Cross-calibration issues can occur when switching from a mission to another or when an instrument switches from a mode to another and this is the reason why we only kept Cluster data when the HIA instrument was under the magnetosphere or magnetosheath mode (l.133).

**12) Since decision trees are easily interpretable, it would be interesting to visualize the boundaries and understand their physical implications.**

We agree with the referee about this statement. However, the problem is currently 8D and such boundaries would then hardly be interpretable. From then on, we have two options that are both not especially the best to provide physical interpretations: projections in specific (e.g. the main ones) 2D features planes that will struggle to give a global vision of this boundary Using the principal components, which will provide a global vision of the boundary between the different classes in a feature space that have no real physical meaning The three different regions could eventually be studied from a massive statistical point of view and this would be the logic aftermath of our massive detection in an upcoming work.

**13) Line 226: it is well-known that the probabilities output of boosted ensemble of decision trees are not well-calibrated.**

The probability calibration of Boosted ensemble of decision trees is indeed well-known

(Niculescu-Mizil et al.) and something we haven't mentioned in the paper, we ensured that in our case, our probabilities were decently calibrated and we attach the calibration curves for each of the three classes.

**14) Line 275: Do Appendixes B and C really have no text?**

Appendixes B and C are just set to represent ROC curves and a 2D histogram of B and Np in the case of Artemis. A small sentence similar to the one in Appendix A will be added to refer to the associated figures.

**15) Finally I suggest to be more specific in the title, by replacing 'Machine Learning' with 'Decision Trees'**

The title could indeed be modified accordingly. Nevertheless, we want to focus on the benefits of applying machine learning to this specific problem rather than focusing on the possible (and yet to be evaluated) specific benefits of Gradient Boosting classifiers against other algorithms. We would then be more eager to keep the current title.

———————————————————

[Figure]

**Fig. 1.** ROC curve obtained with a temporal split for our model trained on THEMIS data

**Fig. 2.** Probability calibration curve for our model trained on THEMIS data

---

## Author Comment (AC2) · 29 Nov 2019

We believe that there is a big misunderstandings about our answer. Please, let us try to clarify the situation.

The referee seems to think that we train a first model with very few points and *automatically* add the positive predictions to the training set of another, etc. We agree with the referee that this methodology would obviously be wrong, as explained in its comments.

This is **not** what is done.

**All labelled points added to the dataset are selected by visual inspection for all**

[Figure]

**spacecraft.** The predictions of intermediate trainings are **just used as a visual suggestion** for labelling the data, to guide the eye. We **manually** build the new training/test set and train a brand new model with our new visually made labels.

The visual guidance the intermediate prediction provides when we label additional data only serves to speed up the data browsing. Indeed, if not perfect, the intermediate predictions are not stupidly wrong either and quite better than random. This prediction is just used to zoom in the data intervals of interest and manually select points we identify as belonging to a given class. Labelling is done also by looking at the spectrograms, as presented in the paper, while this data is not included in the training dataset.

The results presented in the paper are produced with a totally independent, unique, model, trained with the dataset obtained after the whole visual labelling process.

Furthermore, an important part of the paper is dedicated to a **massive prediction on unseen data**. This prediction is presented and is very good. Contrary to many (almost all) studies, we made all codes and predictions available, which can confirm the prediction is good on unseen data. This could not be the case with an algorithm that would be based on only 6 hours of data and successively automatically trained on raw positive inputs. We will clarify this explanation in the revised paper.

---

## Referee Comment (RC3) · Anonymous Referee #2 · 10 Jan 2020

This work presents a Decision Tree based classification scheme for the detection of the various parts of the Magnetoshpere, i.e. the inner magnetosphere the magnetosheath and the solar wind, based on in-situ magnetic field and plasma measurements. The method is built on data from the THEMIS mission and then used (with and without retraining) for other missions as well, such as the Cluster and Double Star. In addition they claim that the same method could potentially be used to separate other regions of the interplanetary space, as is the "Lunar Wake" area demonstrated by using data from the THEMIS/Artemis satellites. The method outperforms simpler classification schemes, such as a typically used threshold based set of criteria and can be utilized to run in a large set of data, thus providing results for the accurate detection of the

Magnetopause and the Bow Shock for a large time period.

The method is highly promising and specifically the comparison of the Gradient Boosted Decision Trees against a typical (and similar) threshold-based approach can be a powerful demonstration of the capabilities of Machine Learning techniques. Unfortunately, the authors do not provide enough details on the method use, while some elements of the construction of the test and training datasets seem somewhat unclear. An updated version of this paper with an expanded Section 2.2 and with a more thorough analysis of the labeling procedure could be fit for publication.

Additional Comments:

Lines 52-55: I would require more detail in the labeling process. What exactly is meant by "successive, eventually corrected, predictions"? The term "training set" is used here, even though the proper training set is defined later, on line 61. Does the final dataset cover the full time range of the 2007-2009 period? Are the authors concerned that their dataset might not be representative of a variety of solar/magnetospheric conditions, since 2007-2009 was near the solar minimum and was a rather quiet time period with regards to geospace activity.

Section 2.2 Algorithm: I would advise the authors to extend this section with more details on the specific way with which they have implemented the method (number of decision trees, cost function, how exactly does the final probability score emerge etc). Providing an entire book as a reference is not particularly helpful and I believe that many readers would be interested in the technical details, since ML is being used in an increasing number of applications these days.

Section 2.3: An additional metric would be welcome here, e.g. the Heidke SS, especially since the AUC scores are pretty close to one another.

Table 1: Since the scores for all three classes are almost perfect, I would expect the mislabeled AUC to be 75% everywhere, but its 70.7% for the Magnetosheath class.

Do you have any suspicions or thoughts on why that might be? Did you also perform 3 different misslabelings to verify this result (as you did with the training-test dataset selection)?

Section 3: Since different satellites carry different instruments with varying sensitivities it would be interesting to see if using some sort of normalization scheme in the data can help the method to yield high scores without re-training.

Also, it is generally advised to use an as-equal-as-possible sample size for all the classes in a dataset. Especially in the Double Star case the Solar Wind category seems significantly under-represented compare to the other two. Have the authors tried to replicate their results with a more balanced dataset?

Section 3.3: Wouldn't a set of Lunar coordinates (selenocentric) be more useful in properly identifying the fourth class? Or alternatively and additional parameter that captures the Moon's Local Time position? Also, I do not see the AUC scores for the fourth category in Table 1.

Lines 227-228: "Events with high probability would then correspond to undoubtful crossings while the events with the lowest probability would be the most likely to be actual crossings". Is this correct or was it meant to be "while the events with the lowest probability would be LESS likely to be actual crossings".

Section 5: It would be very interesting to see the difference in the position of the Magnetopause and the Bow Shock for quiet vs disturbed conditions (e.g. low vs high solar pressure) as predicted by this method and a comparison against an analytical model.

---

## Author Comment (AC3) · 29 Jan 2020

We thank the referee for the numerous comments regarding our methodology and the various suggestions proposed to better our work. Please find below the detailed answers to the referee's questions and comments.

**Lines 52-55: I would require more detail in the labeling process. What exactly is meant by "successive, eventually corrected, predictions"? The term "training set" is used here, even though the proper training set is defined later, on line 61.**

The labels were made by inspecting the data visually and deciding, by selecting intervals, to which class their points belonged to. This requires to zoom in and out many intervals and is thus a long and fastidious process. To make it faster, in particular to zoom in regions of interest, we decided to guide our eyes with the preliminary predictions of a GB classifier trained on a dataset iteratively widened by our labels, plotted over the data.

**Does the final dataset cover the full time range of the 2007-2009 period? Are the authors concerned that their dataset might not be representative of a variety of solar/magnetospheric conditions, since 2007-2009 was near the solar minimum and was a rather quiet time period with regards to geospace activity.**

The final labeled dataset we used for THEMIS data does cover the 2007-2009 period. Past this period, THEMIS B and C became Artemis and this specific case was mentioned in the paper. The concern of the variability due to the solar cycle then concerns the quality of the massive prediction lead on the data provided by THEMIS A, D and E.

Even if the solar cycle induces variability in the physical parameters of the three regions, we will find the same differences between the three different classes on the dayside part of the near-Earth environment:

- The magnetosphere will still be characterized by a low density, a high temperature, a high magnetic field and a plasma almost at rest

- The magnetosheath will still be characterized by a high density, a subsonic ion flow, and a lower magnetic field amplitude than in the magnetosphere

- The solar wind will still be characterized by a supersonic ion flow, a high density being lower to the one in the magnetosheath and a low magnetic field amplitude.

At first order, this physical differences we have between each classes will prevail on the variability induced by the solar cycle. The latter can then be neglected in the specific case of these missions.

**Section 2.2 Algorithm: I would advise the authors to extend this section with more details on the specific way with which they have implemented the method (number of decision trees, cost function, how exactly does the final probability score emerge etc). Providing an entire book as a reference is not particularly helpful and I believe that many readers would be interested in the technical details, since ML is being used in an increasing number of applications these days.**

Gradient boosting algorithms have proven their capability to rapidly deal with complex, eventually imbalanced (Brown et al. (2012) ) classification problems. This is the reason for which we chose this class of algorithms. We computed the method using its python implementation provided by Scikit-learn (Pedregosa et al. (2011) ). The method has been computed with the standard hyperparameters provided by Scikit-learn that is to say:

- 100 decision tree

- The multinominal deviance loss function which is a standard loss function used for multiclass classification

These precisions will be added in the revised manuscript

**Section 2.3: An additional metric would be welcome here, e.g. the Heidke SS, espe- cially since the AUC scores are pretty close to one another.**

We agree with the referee and attach to our answer the evolution of the HSS as a function of the probabilistic decision threshold. For the decision threshold we chose for our massive detection (e.g 0.5), the high value of the HSS we obtain for the three classes confirms the efficiency of our model. This figure will be added to the revised version of the paper where we will also introduce this metric.

**Table 1: Since the scores for all three classes are almost perfect, I would expect the mislabeled AUC to be 75C2 Do you have any suspicions or thoughts on why**

Interactive
comment

**that might be? Did you also perform 3 different misslabelings to verify this result (as you did with the training-test dataset selection)?**

The two only reasons of a data mislabel by a human observer are the confusion of magnetosheath points with either magnetosphere or solar wind points. Consequently, we mislabeled our dataset following this process:

- we selected a fraction of random points of the dataset

- The magnetosphere and the solar wind points were mislabelled as magnetosheath points

- magnetosheath points were randomly mislabelled between the two different classes

This operation has been done for various percentages of mislabelling 10 times each. We attach to our answer the evolution of the AUC for each class with the mislabelling.

Even if the classifier is almost perfect, we do not expect a particular evolution in the AUC for each class apart from a drop in the model performance. From then on, having the same performances for the solar wind and the magnetosphere for this percentages is a coincidence that is seen by the evolution of the AUC for these two classes.

This mislabelling process and the evolution of the AUC with the mislabelling will be added in the annex of the revised paper.

**Section 3: Since different satellites carry different instruments with varying sensitivities it would be interesting to see if using some sort of normalization scheme in the data can help the method to yield high scores without re-training.**

Cluster has a polar orbit while THEMIS and Double Star stay in the equatorial plane. This physical difference in latitude of the different regions traversed by the spacecraft is
the main reason that explains the variability we notice in the data from a mission to another and prevails on the instrumental specificity of each spacecraft. This is especially shown with the score we obtain on Double Star and Cluster without retraining.

A global model that takes into account this orbital variability could be a nice improvement of our models in a future work.

**Also, it is generally advised to use an as-equal-as-possible sample size for all the classes in a dataset. Especially in the Double Star case the Solar Wind category seems significantly under-represented compare to the other two. Have the authors tried to replicate their results with a more balanced dataset?**

Gradient boosting has proven its efficiency to deal with imbalanced dataset (Brown et al. 2012) and this is one of the reason for which we chose this algorithm, this will be precised in the paper.

Additionally, we show in the paper that the model trained on THEMIS data already gets on well with every categories of data measured by Double Star. As the retrain already takes into account the differences between the three classes represented in the THEMIS dataset , the imbalance will have a tiny influence on the model performances that are already good.

**Section 3.3: Wouldn't a set of Lunar coordinates (selenocentric) be more useful in properly identifying the fourth class? Or alternatively and additional parameter that captures the Moon's Local Time position? Also, I do not see the AUC scores for the fourth category in Table 1.**

We use the plasma moments to take into account all of the possible complexity of the underlying nature of the object we are trying to detect, without simplifying a priories coming from modelization

This approach is preferable as long as the different type of signatures we are trying to detect have strong intrinsic properties. Which is not the case for the magnetosheath

and the solar wind in the distant night side. This is why we help the algorithm by giving the position of the spacecraft in this specific case.

We agree we didn't mention the AUC score for the fourth class that we found equal to 0.997. We will mention this score in the revised version of the manuscript.

**Lines 227-228: "Events with high probability would then correspond to undoubtful crossings while the events with the lowest probability would be the most likely to be actual crossings". Is this correct or was it meant to be "while the events with the lowest probability would be LESS likely to be actual crossings".**

We agree with the referee on the omission we made on this sentence. This will be corrected.

**Section 5: It would be very interesting to see the difference in the position of the Magnetopause and the Bow Shock for quiet vs disturbed conditions (e.g. low vs high solar pressure) as predicted by this method and a comparison against an analytical model.**

We agree such study would be interesting and this is why we mention it in the conclusion of our paper. The construction of a data-driven bow shock and magnetopause model is one of the objectives we can have by performing massive crossing detection from in-situ data and this was studied by an intern from our group. The preliminary plots we obtained showed consistency with what we know on the position of the bow shock for varying solar wind dynamic pressure. Nevertheless, this work is very preliminary and would need a specific focus that goes beyond the scope of this paper.
* * *
[Figure]

**Fig. 1.** Heidke Skill Score of our model trained on THEMIS data for varying decision threshold for the three classes

[Figure]

**Fig. 2.** Evolution of the AUC as a function of the percentage of mislabelled points in the dataset. The grey dashed lines indicate the error bars obtained for each 10 repetitions of mislabelled training